# Detection of Forest Windstorm Damages with Multitemporal SAR Data—A Case Study: Finland

Erkki Tomppo [1,2,*], Ghasem Ronoud [1], Oleg Antropov [3], Harri Hytönen [4] and Jaan Praks [1]

1 Department of Electronics and Nanoengineering, Aalto University, P.O. Box 11000, 00076 Aalto, Finland; ghasem.ronoud@aalto.fi (G.R.); jaan.praks@aalto.fi (J.P.)
2 Department of Forest Sciences, University of Helsinki, Latokartanonkaari 7, P.O. Box 27, 00014 Helsinki, Finland
3 VTT Technical Research Centre of Finland, P.O. Box 1000, 00076 Espoo, Finland; oleg.antropov@vtt.fi
4 Finnish Forest Centre, Kauppakatu 19 B, 40100 Jyväskylä, Finland; harri.hytonen@metsakeskus.fi
* Correspondence: erkki.tomppo@aalto.fi or erkki.tomppo@helsinki.fi

**Abstract:** The purpose of this study was to develop methods to localize forest windstorm damages, assess their severity and estimate the total damaged area using space-borne SAR data. The development of the methods is the first step towards an operational system for near-real-time windstorm damage monitoring, with a latency of only a few days after the storm event in the best case. Windstorm detection using SAR data is not trivial, particularly at C-band. It can be expected that a large-area and severe windstorm damage may affect backscatter similar to clear cutting operation, that is, decrease the backscatter intensity, while a small area damage may increase the backscatter of the neighboring area, due to various scattering mechanisms. The remaining debris and temporal variation in the weather conditions and possible freeze–thaw transitions also affect observed backscatter changes. Three candidate windstorm detection methods were suggested, based on the improved k-nn method, multinomial logistic regression and support vector machine classification. The approaches use multitemporal ESA Sentinel-1 C-band SAR data and were evaluated in Southern Finland using wind damage data from the summer 2017, together with 27 Sentinel-1 scenes acquired in 2017 and other geo-referenced data. The stands correctly predicted severity category corresponded to 79% of the number of the stands in the validation data, and already 75% when only one Sentinel-1 scene after the damage was used. Thus, the damaged forests can potentially be localized with proposed tools within less than one week after the storm damage. In this study, the achieved latency was only two days. Our preliminary results also indicate that the damages can be localized even without separate training data.

**Keywords:** boreal forest; windstorm damage; synthetic aperture radar; C-band; Sentinel-1; support vector machine; improved k-NN; genetic algorithm; multinomial logistic regression

## 1. Introduction

### 1.1. Background and Objectives of the Study

Windstorm damages have become more common in the past decades [1,2]. Windstorms cause noticeable large area forest damages in Europe, including Scandinavia and Finland. For example, in southern Sweden, approximately 4.5 million cubic meters of timber was damaged in 1999 in a single storm [3], and in 2005 and 2007 approximately 70 and 12 million cubic meters of timber fell down in similar disastrous events, respectively [4]. The forest area reported to have been cut due to damages was over 30,000 ha on more than 20,000 forest stands in Northern Finland in 2014, and more than 6000 ha in Eastern Finland in July 2020.

A rapid localization of the forest damages and removal of the fallen trees is the key for not only assessing the losses, but also avoiding further damage, caused, e.g., by insects. Severe storms require earlier sanitary cuttings (compared to original forest plans) to prevent

such insect outbreaks. These ad-hoc cuttings naturally increase harvesting and removal costs, cause losses in revenue and lower the future cutting possibilities [5]. The volume of damaged trees in windstorm has exceeded the volume of the normal annual cut in some countries in Europe, e.g., Germany, Poland and Sweden [1,6]. Timely detection and mapping of a damaged forest allows additionally to optimize efforts in clearing potentially blocked roads and damaged power-lines in rural areas.

A common method to localize the damage areas for operational forest regime purposes and obtain a rough overview of the damages has been monitoring with airplane using either visual assessment or optical sensors, e.g., video camera or airborne laser scanner (ALS). Most large-area studies with space-borne data have been conducted using optical satellite instruments [7]. A recent study by Rüetschi et al. [8] presents a summary of several demonstrated approaches in mapping windthrown forest areas. Our further in-depth analysis with SAR data is given in Section 1.2.

A key prerequisite for successful operational forest management after a storm is a rapid, near-real-time localization of the damages. Damages are often large in area wherefore methods using space-borne data are appealing and cost-efficient alternatives. A central requirement is the timely availability of the remotely sensed data. SAR data are the only possibility for rapid monitoring due to their independence of light conditions and cloud cover.

SAR backscatter depends on the forest structure and biomass, the environment and weather conditions such as moisture and temperature and sensor properties. From the current operative SAR satellites, EU Copernicus program's two C-band Sentinel-1 satellites probably have the best potential for rapid monitoring, primarily due to a frequent data acquisition and a free of charge data policy [9]. The only drawback of C-band data in forest application is the low penetration to forest volume due to short wavelength, which could restrict detectability of minor damages. The ALOS PALSAR-2 with L-band SAR, with deeper penetration depth than C-band and fully polarimetric capability, would likely better suit forest applications [10,11], but the operational use is restricted by data availability [10,11]. ESA's coming forest specific P-band BIOMASS mission may provide information for monitoring aboveground biomass and its change over large areas, but will not be operated over Europe [12]. The data availability and price also restrict the usability of high resolution X-band SAR data that could enable spatial texture analysis of SAR backscatter for forest disturbance [10] (a further detailed analysis of possible scenarios is given in Section 1.2) In the future, new satellites and constellations, such as NISAR [13] and ICEYE [14,15] as well as planned DLR TanDEM-L [16] and ESA ROSE-L [17] may improve the situation significantly.

Thus, Sentinel-1 presently and in the near future seems to be the most suitable tool for forest damage assessment in Europe at operational level.

The overall goals of this study were to develop methods to localize the forest windstorm damages, assess the severity and area of damaged forests and quantify the uncertainties in forest damage prediction when using space-borne SAR data.

The detailed objectives were:
1. to study the potential of Sentinel-1 SAR data in localizing the forest windstorm damages;
2. to assess the accuracy of the developed methods; and
3. to assess the time lag from the damage to the damage detection and a ready product.

### 1.2. Windstorm Damage Studies with SAR

Several windstorm studies with SAR are shortly reviewed in this section, as well as studies using airborne SAR instruments. It is expected that the number of SAR-based studies will increase with the increasing data availability.

Green [18] investigated the sensitivity of SAR backscatter to forest windstorm damage gaps using multi-polarization C, L and P band data acquired by the NASA/JPL AIRSAR in August 1991. The study showed that changes in backscatter due to the presence of windstorm damage gaps were evident in each polarization channel used, especially with C-

band HH polarization in a coniferous plantation. It is suggested that backscatter is sensitive not only to the presence but also to the shape and geometry of the windthrow gaps.

Dwyer et al. [19] used ESA's C-band ERS-1/ERS-2 interferometric image pairs and found them to be effective in differentiating between damaged and undamaged forests when the damaged areas were larger than or equal to 2–3 ha. The damage happened in Jura mountain in France in December 1999. Ready software by ESA made fast data processing possible.

Fransson et al. [3] studied the potential of CARABAS-II long wavelength SAR imagery for high spatial resolution mapping of windstorm damage forests. The results of this research show that the backscattering amplitude, at a given stem volume, is considerably higher for windstorm damage thrown forests than for unaffected forests. In addition, the backscattering from fully harvested storm-damaged areas was, as expected, significantly lower than from unaffected stands. These findings imply that VHF SAR imagery has potential for mapping windthrown forests.

Another study by Fransson et al. [4] investigated simulated wind-thrown forest mapping (controlled experiment with felling of trees) using multitemporal ALOS PALSAR (L-band), RADARSAT-2 (C-band) and TerraSAR-X (X-band) imagery. The detection methodology was based on bitemporal change detection and visual interpretation of scenes acquired before and after a simulated windthrow event. Stripmap ALOS PALSAR images were found less suitable for a damage area detection, likely due to a coarse spatial resolution. The windthrown forests were well visible when the RADARSAT-2 and TerraSAR-X HH polarization images were used.

Ulander et al. [20] used space- and airborne SAR data to map windthrown forests in southern Sweden. Analysis of the Space- and Airborne C-band SAR images including Envisat and Radarsat showed that they are unable to detect forest storm damage. The CARABAS VHF-band SAR, on the other hand, showed that these data can detect most storm-damaged forests as well as power lines, and sometimes better than the aerial photographs.

Thiele et al. [21] used TerraSAR-X data and focused first on the border line extraction of forest areas to enables a fast estimation of windthrown areas, whereby the pre-event forest border is derived from multi-spectral data. Second, clean-up operations were monitored in the affected forest areas by applying a change detection operator. They presented a method to extract the border of forest areas by fusing multi-aspect SAR images. They found that this extraction of multi-temporal changes and displacements of the forest border enables a rapid damage estimation, which is very useful to plan first clean-up operations. In addition, their intensity-based change detection showed good results to highlight small areas especially with hard to analyze data, even for human operators.

Eriksson et al. [6] showed that, when trees are felled, the backscattered signal from TerraSAR-X (X-band) increases by about 1.5 dB, while for ALOS PALSAR (L-band) a decrease with the same amount is observed. Radar images with fine spatial resolution also showed shadowing effects that should be possible to use for identification of storm felled forest.

Tanase et al. [22] applied L-band space-borne SAR data to windthrow and insect outbreak detection in temperate forests. The results show that changes in backscatter relate to the damages caused by the wind and insect outbreaks. In this case, an overall accuracy of 69–84% was achieved for the delineation of areas affected by the wind damage. The study showed that L-band space-borne SAR data can be employed over larger areas and ecosystem types in the temperate and boreal regions to delineate and detect damaged areas.

Rüetschi et al. [8] developed a straightforward approach for a rapid windthrow detection in mixed temperate forests using Sentinel-1 C-band VV and VH polarization data. Following radiometric correction of Sentinel-1 scenes acquired approximately 10 days before and 30 days after the storm event, a SAR composite images of before and after the storm were generated. The differences in backscatter before the storm and after the storm in windthrown and in intact forest were studied. A change detection method was developed.

Locations of windthrown areas of a minimum extent of 0.5 ha was suggested. The detection was based on user-defined parameters. While the results from the independent study area in Germany indicate that the method is very promising for detecting areal windthrow with a producer's accuracy of 0.88, its performance was less satisfactory at detecting scattered windthrown trees. Moreover, the rate of false positives was low, with a user's accuracy of 0.85 for (combined) areal and scattered windthrown areas. These results underscore that C-band backscatter data have a great potential to rapidly detect the locations of windthrow in mixed temperate forests within approximately two weeks after a storm event.

Other methods potentially suitable for mapping windthrown forests with SAR data include approaches demonstrated in other studies of natural and/or anthropogenic forest disturbance. These include mapping snow-damaged forest areas [23], monitoring selective logging and thinning operations in boreal and tropical forest biomes [11,24–28], forest clear cutting and other forest changes [29–34].

A common observation is that, while at L-band direct pixel-wise (or area-based) change detection using averaged-backscatter can be attempted, due to a better sensitivity to forest structure and volume, this does not really work at shorter wavelength such as C-band. Especially in the absence of fully polarimetric SAR capability. At C-band, texture analysis/extraction and subsequent image segmentation should be attempted after speckle is reduced (e.g., using image aggregation of scenes acquired before and after the forest disturbance event). At C-band, felling of trees does not change strongly the total backscatter, since the needles and smaller branches still create "bright enough" random-volume layer. Thus, texture from shadows in standing and fallen trees is the key feature to rely upon in the analysis. At even shorter wavelength and even higher resolution, such as X-band, texture analysis becomes the central way to proceed with the change detection, in addition to single pass interferometry with X-band data. Interestingly, most of the studies cited above rely on some kind of bitemporal change detection (even if "before" and "after" scenes are aggregated in two composite images). This does not really allow analyzing the added value of incorporating additional scenes of temporal dimension into the analysis. However, the idea of using textural features, even at stand level, and follow-up image segmentation appears most fruitful and is adopted and elaborated in our further analysis and methodology development.

## 2. Material

### 2.1. Test Site

The study area was selected in a collaboration with the Finnish Forest Centre. It is a forested landscape in Southern Finland in which a severe windstorm damage occurred on 12 August 2017. The fastest speed of the wind in the inland was near 30 m/s and on the sea outside the capital 32 m/s [35]. The area reaches from Helsinki capital region towards the towns Kouvola and Lappeenranta east and northeast of Helsinki region (Figure 1). The area of the forestry land in the study area, covered by Sentinel-1 scenes, is 830,000 ha. Forestry land includes three land categories: (1) forest land; (2) poorly productive forest land; and (3) unproductive land [36]. The two commonest stand level dominant tree species are Norway spruce (*Picea abies* Karst. L.) and Scots pine (*Pinus sylvestris* L.) (see also Tables 1 and 2).

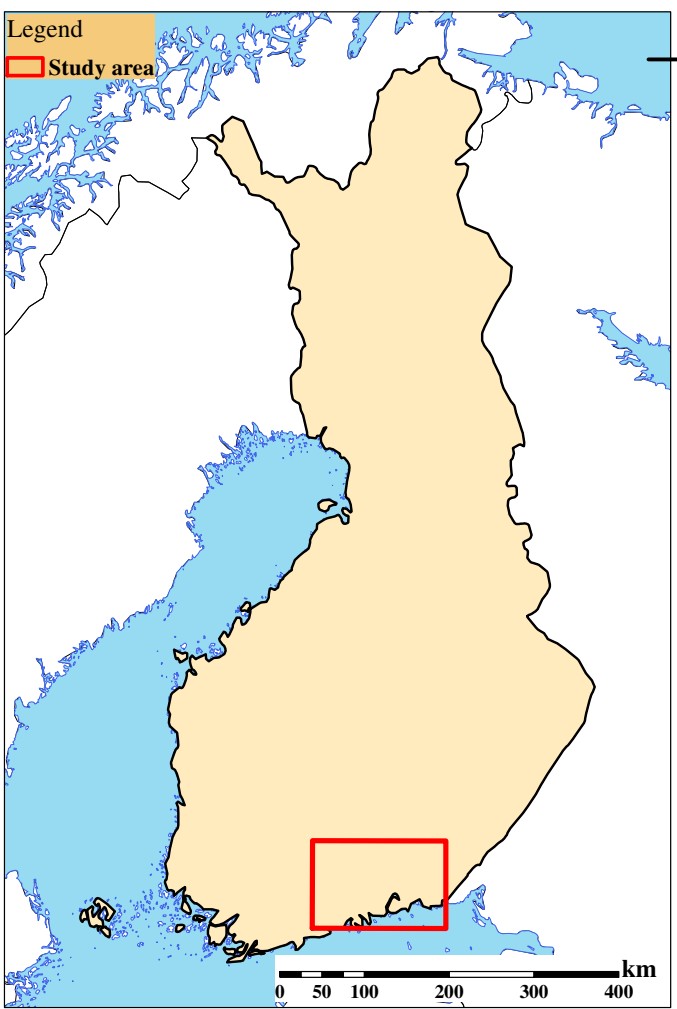

**Figure 1.** The location of the study area in Southern Finland.

*2.2. Field Data*

The training data and validation data were also selected in a collaboration with the Finnish Forest Centre and extracted from their forest database. The harvesting operation in Finnish forests presume an advance announcement and acceptance by the Forest Centre wherefore the Forest Centre has good overview of stands with wind damage but only after a longer reporting period. Two types of stands were selected for study: (a) stands in which harvesting had been planned to be carried out due to the windstorm damage; and (b) stands in which harvesting had not been reported. We call these stands damaged and non-damaged, respectively, in the following analysis. The total number of stands in reference database falling inside of all images was 977 after screening checks and removal of some doubtful stands, e.g., when stand characteristics seemed to be out-of-date in 2017. The number of damaged stand records was 313 and the number of non-damaged stand records 664. From the damaged stands, 195 stands were severely damaged and the rest slightly damaged. The severity category is assessed in the field by the forestry experts in charge. A severe damage presumes stand regeneration, while slight damage requires only removal of fallen or broken trees. The areas and growing stock characteristics are shown in Table 1 and similar statistics when the volume of growing stock is larger than 75 m$^3$/ha in Table 2.

Damaged and non-damaged stands displayed on Sentinel-1 scene from 19 August 2017, VV polarization and VH polarization are shown in Figure 2 (see also the zoomed figures with stand boundaries in Figures 3 and 4).

**Table 1.** Average stand-level areas and forest characteristics in training data, separately for damaged and non-damaged stands.

| | Damaged Stands | | | Non-Damaged Stands | | |
|---|---|---|---|---|---|---|
| | 313 Stands | | | 664 Stands | | |
| Characteristics | Minimum | Average | Maximum | Minimum | Average | Maximum |
| Area, ha | 0.05 | 1.89 | 33.96 | 0.03 | 0.91 | 7.24 |
| Diameter, cm | 0.00 | 19.57 | 29.25 | 0.00 | 17.79 | 27.79 |
| Height, dm | 2.00 | 169.32 | 253.00 | 0.78 | 159.98 | 247.35 |
| Age, years | 1.00 | 56.20 | 96.61 | 0.63 | 49.27 | 98.92 |
| Basal area, $m^2$/ha | 0.00 | 18.79 | 30.00 | 0.00 | 18.27 | 30.23 |
| Volume, $m^3$/ha | 0.00 | 167.60 | 337.83 | 0.71 | 159.75 | 330.87 |
| Volume, pine | 0.00 | 52.92 | 145.50 | 0.71 | 41.13 | 182.43 |
| Volume, spruce | 0.00 | 84.31 | 304.67 | 0.00 | 68.43 | 284.36 |
| Volume, birch spp. | 0.00 | 21.89 | 110.80 | 0.00 | 34.87 | 107.63 |
| Volume, other br. l. | 0.00 | 8.48 | 103.00 | 0.00 | 15.32 | 67.60 |

**Table 2.** Average stand-level areas and forest characteristics in training data, separately for damaged and non-damaged stands with volume of growing stock greater than 75 $m^3$/ha.

| | Damaged Stands | | | Non-Damaged Stands | | |
|---|---|---|---|---|---|---|
| | 281 Stands | | | 631 Stands | | |
| Characteristics | Minimum | Average | Maximum | Minimum | Average | Maximum |
| Area, ha | 0.05 | 1.89 | 33.96 | 0.03 | 0.91 | 7.24 |
| Diameter, cm | 9.17 | 20.97 | 29.25 | 6.28 | 18.38 | 27.79 |
| Height, dm | 87.83 | 181.34 | 253.00 | 64.49 | 165.06 | 247.35 |
| Age, years | 22.17 | 60.11 | 96.61 | 17.92 | 50.84 | 98.92 |
| Basal area, $m^2$/ha | 7.60 | 20.3 | 30.00 | 8.57 | 18.92 | 30.23 |
| Volume, $m^3$/ha | 77.88 | 181.4 | 337.83 | 76.89 | 165.62 | 330.87 |
| Volume, pine | 5.00 | 57.06 | 145.50 | 1.27 | 42.77 | 182.43 |
| Volume, spruce | 5.54 | 92.17 | 304.67 | 0.03 | 71.35 | 284.36 |
| Volume, birch spp. | 1.00 | 23.27 | 110.80 | 1.70 | 35.88 | 107.63 |
| Volume, other br. l. | 0.00 | 8.88 | 103.00 | 0.00 | 15.62 | 67.60 |

Selection of the Training Data and Validation Data

The field observation data were ordered based on the east and north coordinates and split into training and validation data using a systematic sampling. The goal was a representative variation both in training data and validation data. Three-fourths of the observations were selected into the training data and the remaining one-fourth into the validation data. The entire data were 977 stands, of which 732 were used as the training samples and the other 245 as validation samples (Table 3).

Windstorm damages usually occur in more advanced stands: in thinning forests stands or mature forests stands. The analyses were therefore done also with the forest stands in which the volume of the growing stock on the basis of the Finnish multi-source national forest inventory (MS-NFI) was larger than 75 $m^3$/ha, a value leaving out young forests (Table 3) (see Section 2.4 and the work of Tomppo et al. [36]). The number of those stands was 912 from, of which 683 stands belonged to the training data and 229 stands to the validation data.

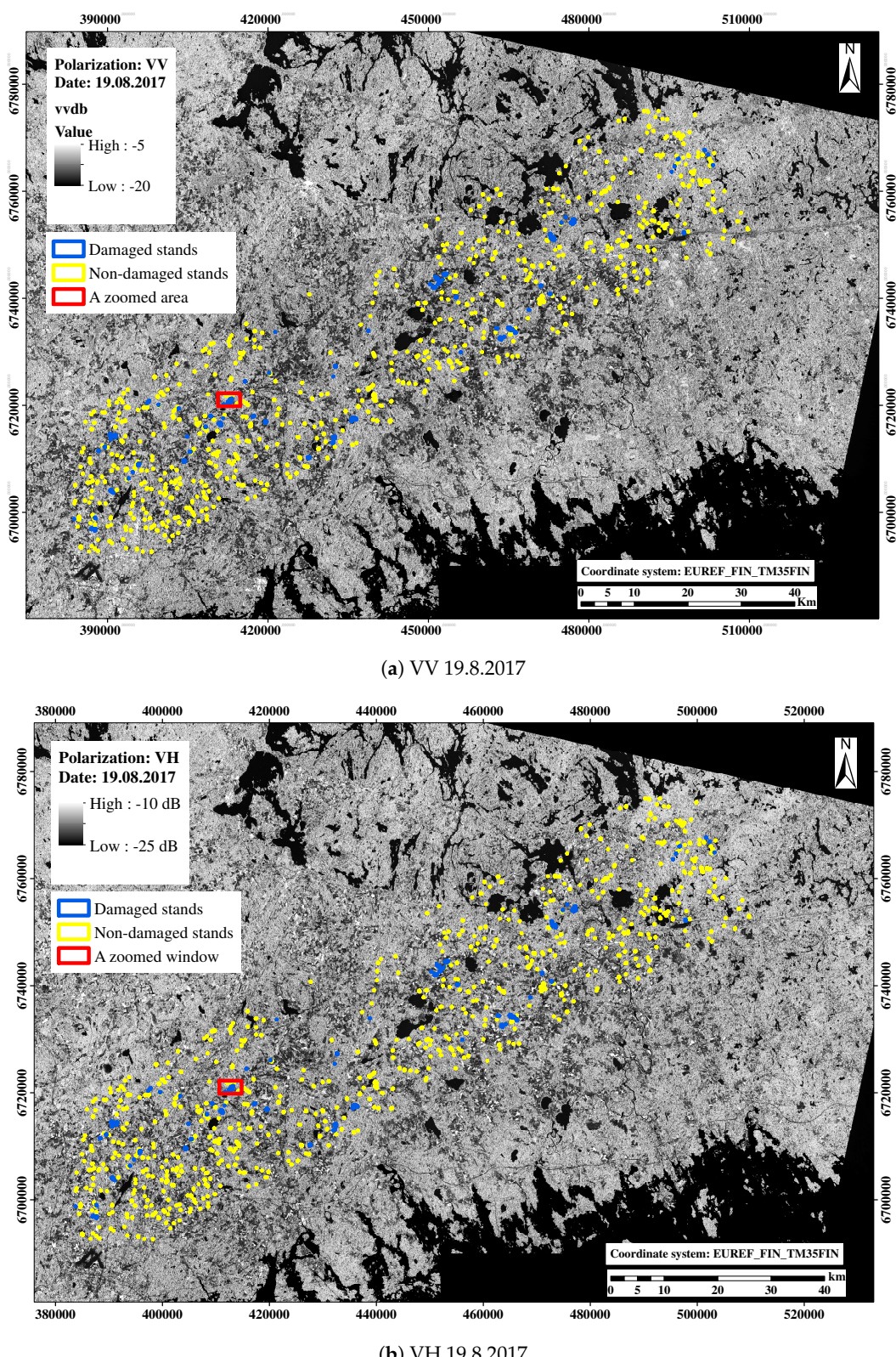

(**a**) VV 19.8.2017

(**b**) VH 19.8.2017

**Figure 2.** Damaged and non-damaged stands displayed on Sentinel-1 from 19 August 2017: VV polarization (**a**); and VH polarization (**b**) (EPSG:3067).

**Table 3.** The numbers of forest stands in the training data and validation data classified also based on the severity of the damage in the entire dataset and when the volume of growing stock is larger than 75 m$^3$/ha.

| Data | Entire Dataset | | | | Volume > 75 m$^3$/ha | | | |
|---|---|---|---|---|---|---|---|---|
| | Damage | | | | Damage | | | |
| | No | Severe | Slight | Total | No | Severe | Slight | Total |
| Training | 492 | 142 | 98 | 732 | 474 | 120 | 89 | 683 |
| Validation | 172 | 49 | 24 | 245 | 57 | 48 | 24 | 229 |
| Total | 664 | 191 | 122 | 977 | 631 | 168 | 113 | 912 |

*2.3. Sentinel-1 SAR Data*

In total, 40 Sentinel-1 GRD (ground range detected) images, Level-1 data with VV and VH polarizations, were downloaded from ESA Open Access Hub. Some of these images covered the test site only partly. In total, 27 image layers were constructed (mosaiced) from the images. The images were acquired in interferometric wide (IW) mode between 4 January and 25 September 2017. Multi-temporal data are necessary for change detection, but aggregating the data also reduces the effect of the random scattering (speckle) on the estimates and error estimates. It makes it possible to utilize the variation of the data acquisition conditions in the estimation using multifaceted information. The dates of the images are shown in Table 4.

The pixel spacing of orthorectified scenes was set to 10 m. The local digital elevation model (DEM) available from National land Survey was used (see Section 2.5). Scenes were aggregated in azimuth and range to obtain images with pixel dimensions approximately corresponding to the 10 m grid spacing. Bi-linear interpolation method was used for resampling in connection with the orthorectification. Radiometric normalization of intensity was done using a projected pixel area-based approach to minimize the effect of the topography. The scenes with a pixel size of about 13.5 m were further re-projected to the ERTS89/ETRS-TM35FIN projection (EPSG:3067) and resampled to a final pixel size of 10 m.

In total, 27 Sentinel-1 mosaics were constructed from the 40 original Sentinel-1 scenes and thus the final data stack includes 54 backscatter intensities layers of VH and VV polarizations. Examples of intensity variations before and after the damages are shown in Figures 3 and 4 with stand boundaries displayed on the intensities.

**Table 4.** The 27 Sentinel-1 data mosaics from 2017 used and their acquisition dates.

| Mosaic | Date | Mosaic | Date | Mosaic | Date |
|---|---|---|---|---|---|
| 1 | 4 January | 10 | 14 July | 19 | 20 August |
| 2 | 16 January | 11 | 15 July | 20 | 1 September |
| 3 | 28 January | 12 | 21 July | 21 | 7 September |
| 4 | 9 February | 13 | 26 July | 22 | 12 September |
| 5 | 21 February | 14 | 2 August | 23 | 13 September |
| 6 | 2 July | 15 | 7 August | 24 | 18 September |
| 7 | 3 July | 16 | 8 August | 25 | 19 September |
| 8 | 8 July | 17 | 14 August | 26 | 24 September |
| 9 | 9 July | 18 | 19 August | 27 | 25 September |

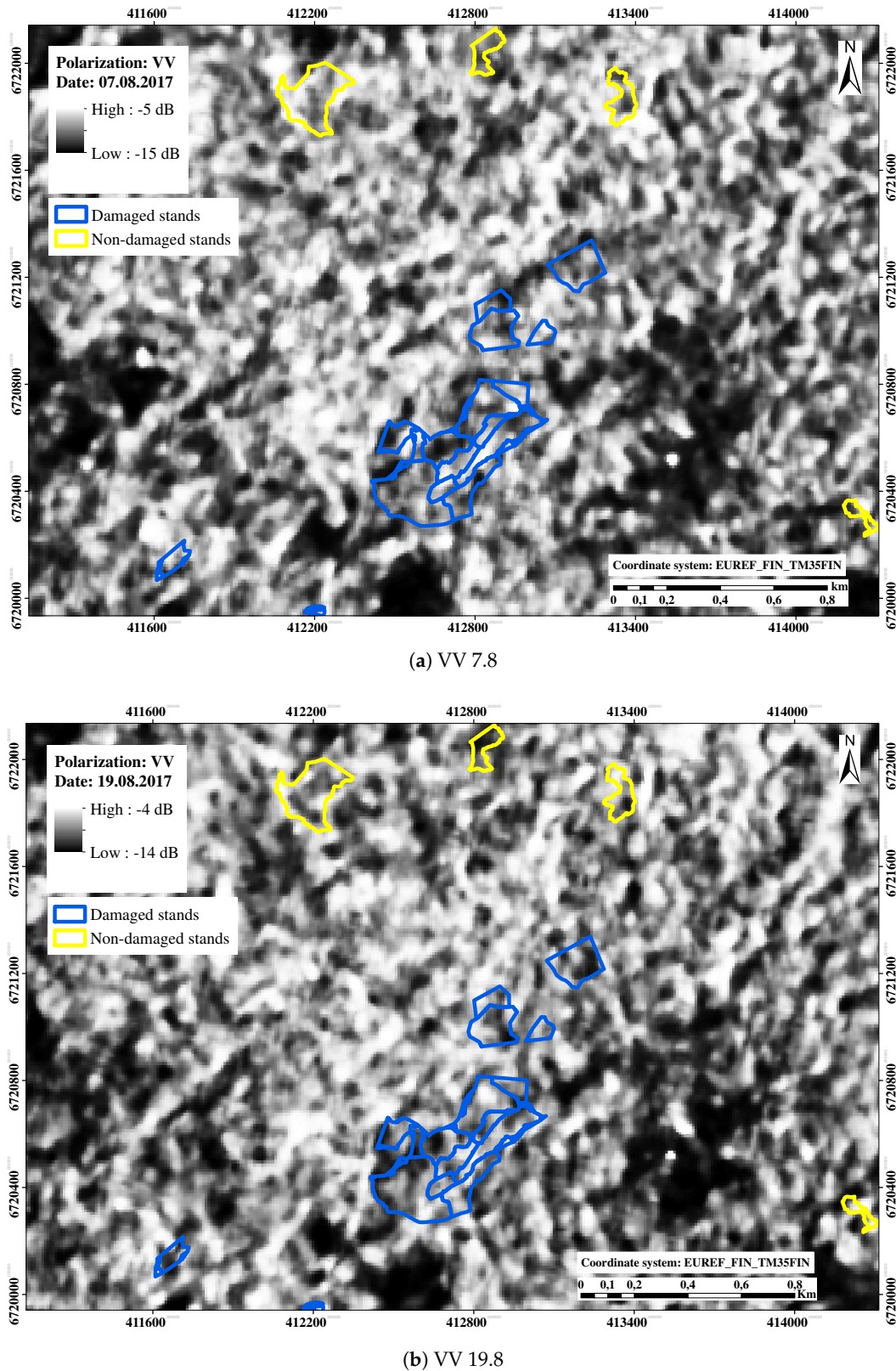

**Figure 3.** Examples of damaged and non-damaged stands displayed on Sentinel-1 scenes: (**a**) before windstorm damage, 7 August 2017; and (**b**) after windstorm damage, 19 August 2017 (**b**) (VV polarization, EPSG:3067).

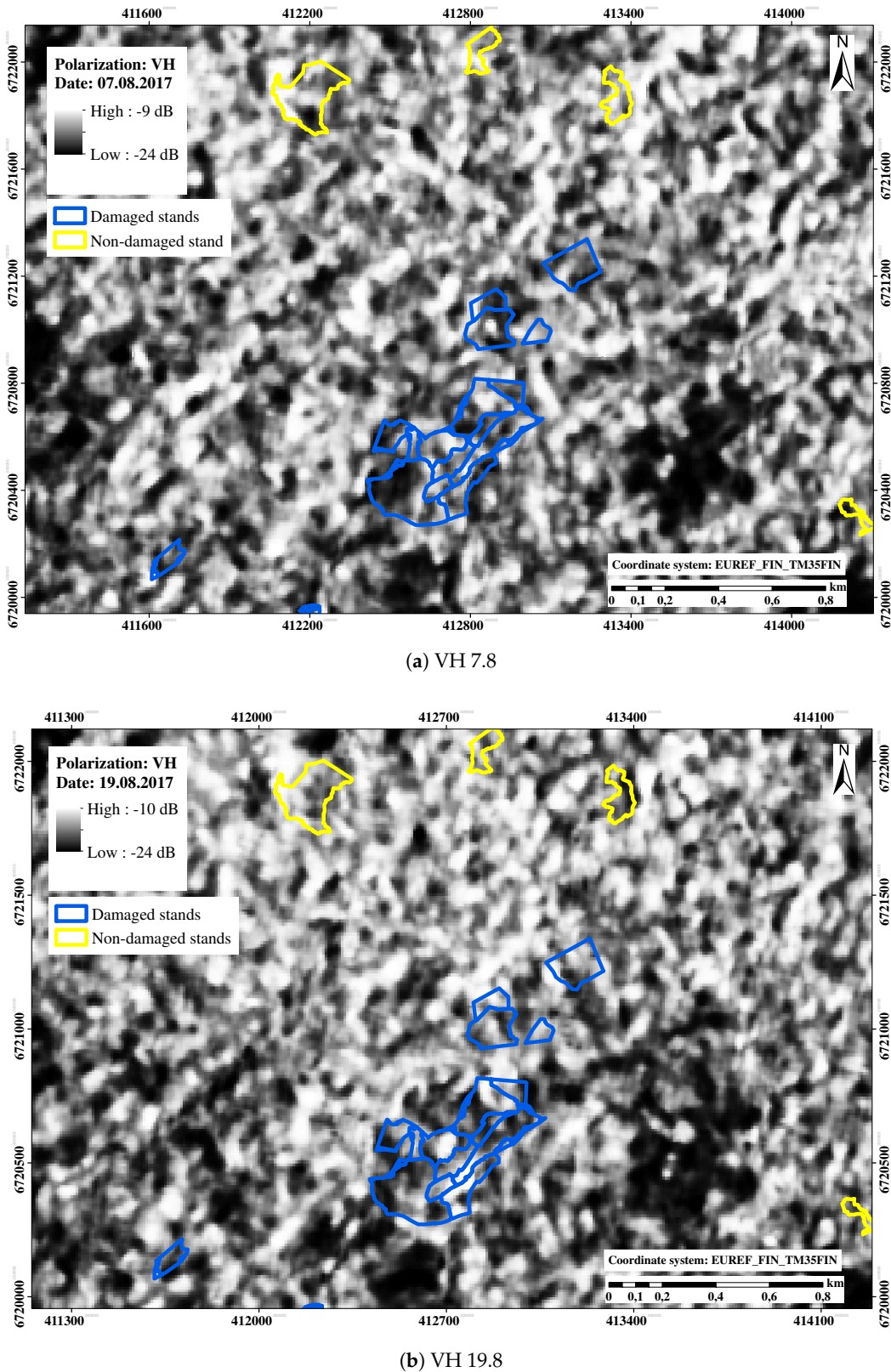

(**a**) VH 7.8

(**b**) VH 19.8

**Figure 4.** Examples of damaged and non-damaged stands displayed on Sentinel-1 scenes: (**a**) before windstorm damage, 7 August 2017; and (**b**) after windstorm damage, 19 August 2017 (**b**) (VH polarization, EPSG:3067).

### 2.4. Multi-Source National Forest Inventory Data

The raster form data from the Finnish multi-source national forest inventory (MS-NFI) were used as additional information in estimating the models to predict the windstorm damages, their severity and uncertainty [36,37]. The data have been projected to correspond to the 31 July 2017 situation and cover all forest ownership groups [38].

The following variables were used in the analyses: mean diameter of the trees, mean height of the trees, mean age of the trees and basal area of trees as well as the volume of the growing stock by tree species groups. The groups were Scots pine (*Pinus sylvestris* L.), Norway spruce (*Picea abies* Karst. L.), birch (*Betula* spp) and other broad leaved trees, mainly aspen (*Populus tremula* L.) and alder (*Alnus* spp.). The first three variables were calculated from the tree-level field measurements as weighted averages, the weight being the basal area of trees. A similar method was used when calculating stand-level averages from pixel-level estimates or measurements. A variation of volume of the growing stock is shown in Figure 5 and a zoom from a sub-area in Figure 6.

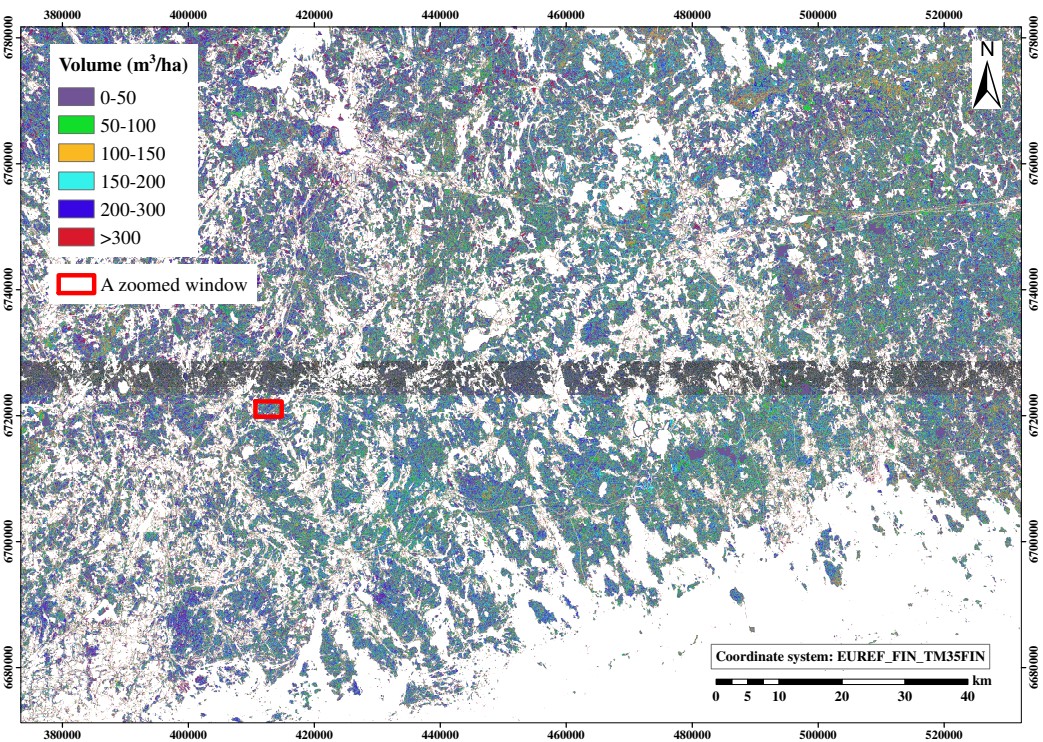

**Figure 5.** Volume of the growing stock on 31 July 2017 on the study area based on MS-NFI (EPS:3067).

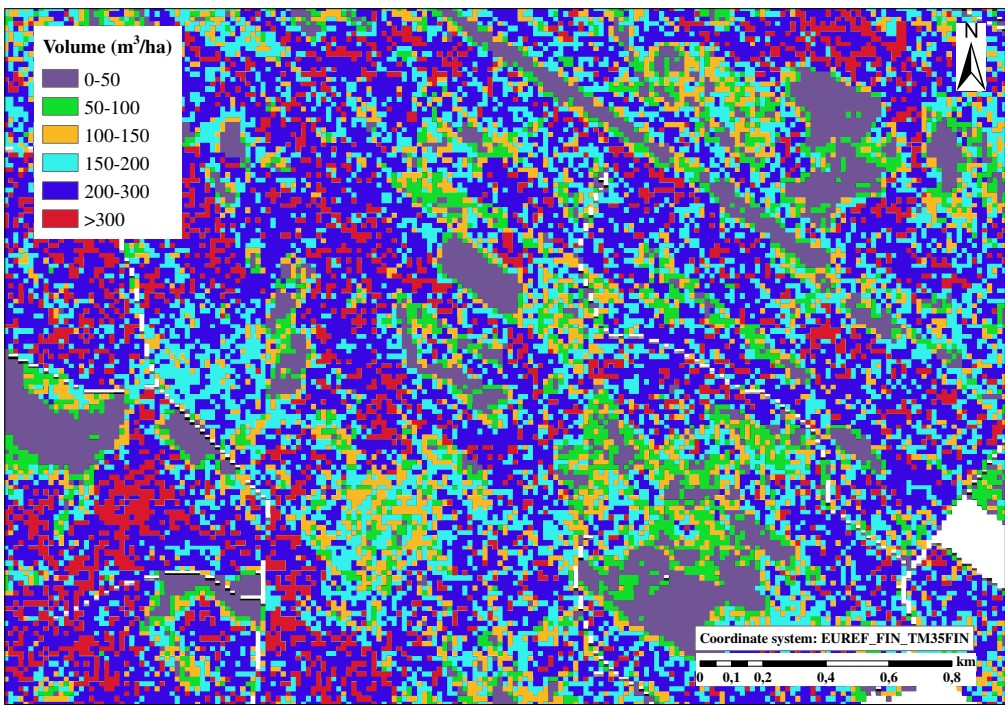

**Figure 6.** Volume of the growing stock on 31 July 2017 on the sub-area of the study area based on multi-source national forest inventory (EPS:3067).

### 2.5. Other Geo-Referenced Datasets

A digital elevation model from Land Survey of Finland was used in orthorectification and radiometric normalization of SAR images, as well as to calculate the average elevation, slope and aspect for each stand and pixel. The original pixel size is 10 m × 10 m and elevation resolution 10 cm [39].

The reason for also using these datasets as explanatory variables in the models is our assumption that the windstorm damages vary also by the aspect and steepness of the slopes of the hills.

## 3. Methods

Three different classification methods, the improved k-NN (ik-NN), multinomial logistic regression (MLR) and support vector machine classifier (SVM), were tailored for windstorm damage detection to reach a desired detection accuracy level. The observations units in the models were forest stand level averages of the Sentinel-1 intensities or other stand level quantities of the input variables. Forest patches identified and derived using a segmentation algorithm were tentatively tested as optional observation units. The reason was that the damages do not necessarily follow the stand boundaries given by the Forest Centre. The segmentation-based units were tested only with the SVM classifier. Figure 7 shows the logic and processing phases of the classification models training and windstorm-map production. The methods are described in detail in the following sections.

The main software tools used for analyses were as follows: (1) SNAP software by European Space Agency [40] was used for Sentinel-1 image pre-processing; (2) GDAL [41] was used for other image raster data pre-processing; (3) statistical computing package R [42] with own codes was used for other data handling including the field data, as well as for MLR, SVM and statistical analyses (Sections 3.4–3.6); and (4) own algorithms for ik-NN (Section 3.3), segmentation and adaptive filtering, written mainly in GNU Fortran [43] (Section 3.2).

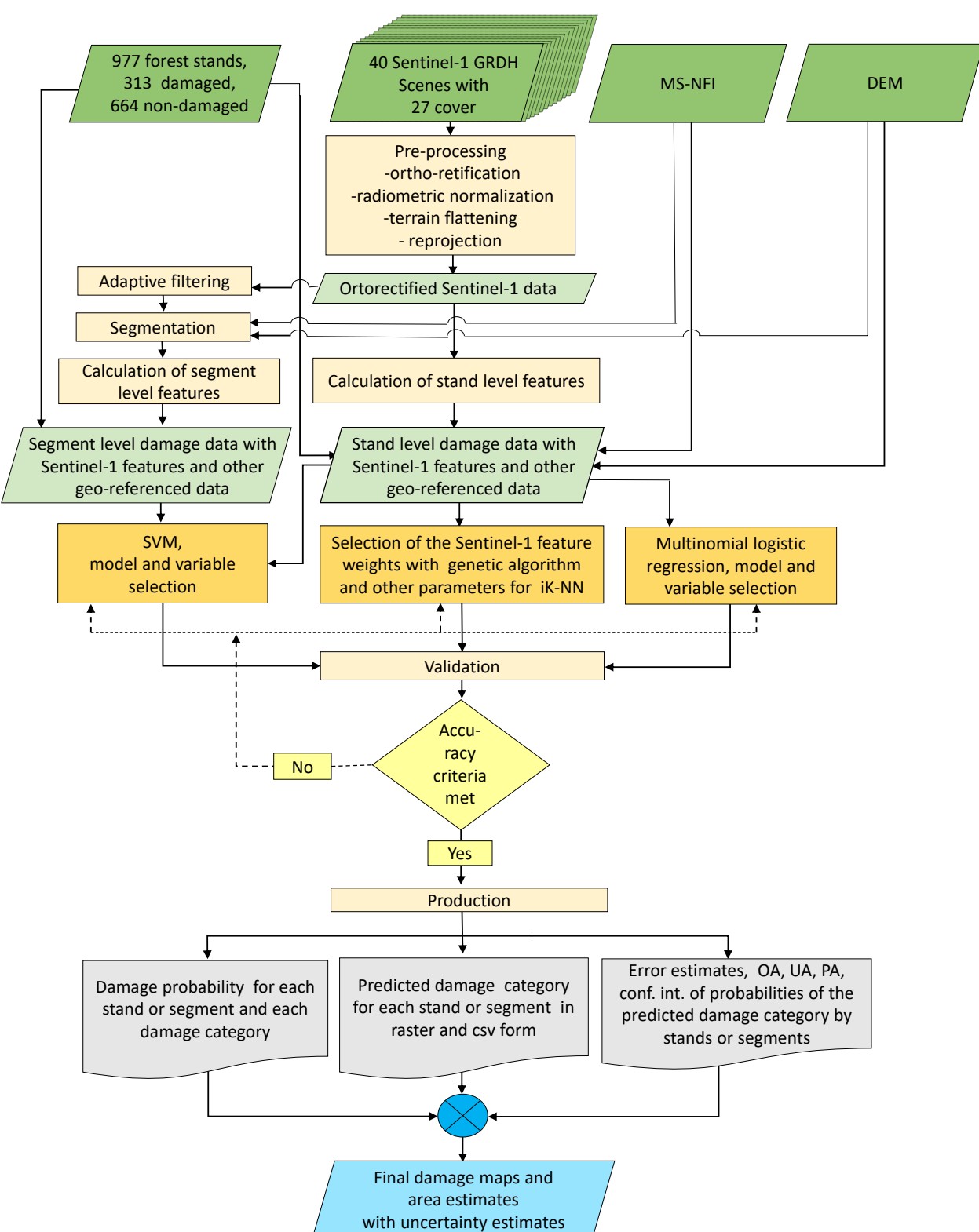

**Figure 7.** The flowchart of the windstorm detection methodology using Sentinel-1 time series: model training and damage-map production phases.

### 3.1. SAR Metrics Used in the Damage Assessment

The basic SAR metrics were stand level variables calculated from pixel level variables for both polarizations. Using stand level metrics instead of pixel level metrics reduces the the speckle effect and the radiometric variation within homogeneous stands. From the pixel level intensities ($I_{s,i}^{k,p}$), the following features were calculated for each stand $s$, for both polarizations $p$ and for each image $k$:

(a) averages,

$$10 \log_{10} \overline{I_s^{k,p}} = 10 \log_{10} \frac{\sum_{i=1}^{n_s} I_{s,i}^{k,p}}{n_s}, k = 1, ..., 27, p \in \{\text{VV}, \text{VH}\}, \tag{1}$$

where $n_s$ is the number of the pixels on stand $s$; (b) standard deviations

$$\sqrt{\sum_{i=1}^{n_s} (I_{s,i}^{k,p} - \overline{I_s^{k,p}})^2 / (n_s - 1)}, k = 1, ..., 27, p \in \{\text{VH}, \text{VV}\}; \tag{2}$$

and (c) intensity-ratios

$$1/n_s \sum_{i=1}^{n_s} I_{s,i}^{k_1,p} / I_{s,i}^{k_2,p}, k_1 = 1 ..., 26 \text{ and } k_2 = 2, ..., 27. \tag{3}$$

These predictor variables, stand-averaged intensities (Equation (1)), standard deviations (Equation (2)) and different ratio features (Equation (3)), were evaluated and, optionally, instead of the averages, also median and mode stand level intensities.

The training data and the validation data and their 'stand' boundaries concerning the damages consisted of the cutting reports rather than the forest patches really damaged by the storm. A natural question is whether it is relevant to use SAR features calculated from the areas reported to be cut because the cutting area can be larger than the damaged area. The characteristics such as median and those calculated from the quantiles SAR features inside the cutting stands were also tested, in addition to the differences of the features from SAR data before and after the damage. Median is not as sensitive to the outliers as is, e.g., the average.

### 3.2. Segmentation-Based Observation Units

The areal units provided by the Finnish Forest Centre represent forest areas to be treated by sanitary cuts due to the windstorm damage. They may be larger than the real damage area. We tested whether it was possible to delineate the entire forest area into sub-areas of which a part of the sub-areas are changed due to the damage. A common practice is to use segmentation for the delineation of the area of interest. The input for segmentation should include the information of the changes. A speckle removal before segmentation could improve the quality of segmentation.

#### 3.2.1. Adaptive Filtering

An adaptive edge-preserving filtering in further stand-level processing was tested. An own heuristic algorithm was implemented and used. It employs a set of alternative image windows (kernels) and selects the one with the smallest variance. The windows were selected inside a window of $5 \times 5$ pixels. The line and column coordinates for a set of 26 different windows are given in Table 5 in the groups of three pixels. For example, $(-2, -1, 0)$ $(2, 1, 0)$ means the set of pixels (line, col) $(-2, 2)$, $(-2, 1)$, $(-2, 0)$, $(-1, 2)$, $(-1, 1)$, $(-1, 0)$, $(0, 2)$, $(0, 1)$ and $(0, 0)$, the center pixel of the window, that is the pixel for which the average value is calculated and attached. Usually 3–5 consecutive runs, depending on data, are needed to identify homogeneous forest patches.

**Table 5.** The 26 image widows (W) for adaptive filtering.

| W | $\text{lin}_{1,2,3}\ \text{col}_{1,2,3}$ | W | $\text{lin}_{1,2,3}\ \text{col}_{1,2,3}$ | W | $\text{lin}_{1,2,3}\ \text{col}_{1,2,3}$ |
|---|---|---|---|---|---|
| 1 | (−2, −1, 0) (0, 0, 0) | 2 | (−2, −1, 0) (1, 0, 0) | 3 | (−2, −1, 0) (1, 1, 0) |
| 4 | (−2, −1, 0) (2, 1, 0) | 5 | (−1, −1, 0) (1, 2, 0) | 6 | (−1, 0, 0) (2, 0, 1) |
| 7 | (0, 0, 0) (0, 1, 2) | 8 | (0, 0, 1) (0, 1, 2) | 9 | (0, 1, 1) (0, 1, 2) |
| 10 | (0, 1, 2) (0, 1, 2) | 11 | (0, 1, 2) (0, 1, 1) | 12 | (0, 1, 2) (0, 0, 1) |
| 13 | (0, 1, 2) (0, 0, 0) | 14 | (0, 1, 2) (0, 0, −1) | 15 | (0, 1, 2) (0, −1, −1) |
| 16 | (0, 1, 2) (0, −1, −2) | 17 | (0, 1, 1) (0, −2, −1) | 18 | (0, 0, 1) (−1, 0, −2) |
| 19 | (0, 0, 0) (−2, −1, 0) | 20 | (−1, 0, 0) (−2, −1, 0) | 21 | (−1, −1, 0) (−2, −1, 0) |
| 22 | (−2, −1, 0) (−2, −1, 0) | 23 | (−2, −1, 0) (−1, −1, 0) | 24 | (−2, −1, 0) (−1, 0, 0) |
| 25 | (0, 0, 0) (−1, 0, 1) | 26 | (−1, 0, 1) (0, 0, 0) | | |

The window with the smallest variance was selected and the average value was attached to the center, that is, the pixel in question. The windows allow the detection of narrow linear form structures in the target and preserves those structures.

### 3.2.2. Segmentation Algorithm

We decided to test whether an algorithm implemented by us could be fine-tuned just for the damage detection with SAR data. The directed tree algorithm by Narendra and Goldberg [44] was modified to use SAR data (see also [45,46]). In this simple test, we used only the difference of the intensities of two scenes, one before and one after the damage and separately VV and VH polarizations.

The algorithm utilizes an edge gradient calculated from an edge image. Any edge operator can be used to construct the edge image. Simplified steps of the segmentation algorithm are as follows. (1) Plateau points are calculated using the inverted edge image and a selected sensitivity (threshold) parameter $\epsilon$. (2) For each pixel $(i, j)$ that is not a plateau point, a parent is determined. The parent of $(i, j)$, $P(i, j)$, is the neighbor that gives the highest positive value of the inverted edge gradient among the neighbors of $(i, j)$. Ties are resolved arbitrarily. If no such neighbor exists, $(i, j)$ has no parent and is therefore a root node. The parents of each plateau points are thus determined. All points on a uni-modal plateau will belong to the same directed tree with one point on the plateau that will be called a root. (3) Once the parent of each point is determined, the points can be labeled by the directed trees (segment) they belong to. The root pixels are first labeled. (4) Once the roots have been labeled, the label of each pixel is determined by tracing of the chains to link the pixels to their root pixels (for details, see [44]). The final segmentation result does not depend on the processing order of the image.

### 3.3. ik-NN Method in Storm Damage Recognition

The well-known k-NN estimation method was tailored for and employed in storm damage recognition. The weights for the features were calculated with a genetic algorithm [36,47] and its variant for categorical variables [48]. This k-NN method is called the ik-NN method (improved k-NN) here. The advantage of the ik-NN method is the weighting of the explanatory variables based on their importance in prediction and thus smaller prediction errors compared to the ordinary k-NN method, wherefore it is called improved. Other advantages of the k-NN method are that all variables can be estimated simultaneously. It preserves thus the natural dependencies of the variables in the estimates, e.g., among stand age, mean height, mean diameter of the trees and the volume of the growing stock. It is non-parametric, and no model is needed. When the weights for training observations are collected for the calculation units, it also avoids a tendency towards the mean in the areal level estimates that is typical for many other methods (see, e.g., [36]). The k-NN estimation method became popular in forest applications when it was taken into into the operational Finnish multi-source forest inventory (e.g., [49,50]). It is very well suited for calculation of the areal level estimates.

Let us recall the main features of the ik-NN estimation with the genetic algorithm in the feature weighting. Denote the $k$ nearest feasible stands by $i_1(p), \ldots, i_k(p)$ when the distance is calculated in the feature space. The weight $w_{i,p}$ of stand $i$ to stand $p$ is defined as

$$
w_{i,p} = \frac{1}{d_{p_i,p}^t} \Bigg/ \sum_{j \in \{i_1(p),\ldots,i_k(p)\}} \frac{1}{d_{p_j,p}^t}, \quad \text{if and only if } i \in \{i_1(p),\ldots,i_k(p)\}
$$
$$
= 0 \quad \text{otherwise.}
$$
(4)

The value of $k$ was fixed to be 5 after preliminary tests using the overall accuracy as the criterion. The distance weighting power $t$ is a real number, usually $t \in [0,2]$. The value $t=1$ was used here. A small quantity, greater than zero, is added to $d$ when $d = 0$ and $i \in \{i_1(p),\ldots,i_k(p)\}$.

The distance metric $d$ employed was

$$
d_{p_j,p}^2 = \sum_{l=1}^{n_f} \omega_l^2 (f_{l,p_j} - f_{l,p})^2,
$$
(5)

where $f_{l,p}$ is the $l$th SAR feature variable of stand $p$, $f_{l,p_j}$ is the $l$th SAR feature variable of the nearest neighbor $j$ of stand $p$, $n_f$ the number of SAR feature variables and $\omega_l$ the weight for the $l$th SAR feature variable.

The values of the elements $\omega_l$ of the weight vector $\boldsymbol{\omega}$ were selected with a genetic algorithm. The details of the genetic algorithm employed are given in [47] and the modification to categorical variables in [48].

The fitness function for the categorical variables to be minimized with respect to $\boldsymbol{\omega}$ vector was

$$
f[\boldsymbol{\omega}, \boldsymbol{\gamma}, B(\mathbf{X})] = \sum_{j=1}^{n_m} \gamma_j [1 - B_j(\mathbf{X}_j)],
$$
(6)

where $\gamma_j > 0$ is a user defined coefficient, $\mathbf{X}_j$ an error matrix, $B_j$ is the accuracy measure with response variable $j$ whose classes are to be predicted, $n_m$ is the number of response variables to be considered in the optimization procedure and $\boldsymbol{\omega}$ is the weight vector to be optimized (Equation (5)). The number generations in the genetic algorithm optimization was selected to be 40 after the tests.

For categorical variables, the mode or median of the predicted classes for the nearest neighbors can be used as a prediction instead of a weighted average as is used for continuous variables. For this study, the mode gave more accurate results than the median, consistent with the earlier investigations [48]. The predicted category is the category that has the greatest sum of the weights, $\omega_{i,p}$, when summed up by classes over the k nearest neighbors. In theory, equal sums are very rare when real value weights are used; in fact, the probability is zero if rounding is not considered. In cases of equal sums for two or more classes, one class is selected randomly from among those with the greatest sum. This method was used for predicting the categorical variable obtaining the values, the value being the damage category.

### 3.4. Multinomial Logistic Regression Method

Multinomial logistic regression was tested as one optional estimation method. The probability of the damage category $k$ on stand $p$ was estimated using the model

$$
\text{P(damage category on stand } p = k | \mathbf{x}_p) = \frac{e^{\beta_k \mathbf{x}_p}}{1 + \sum_{l=1}^{L-1} e^{\beta_k \mathbf{x}_p}}, k = 1, \ldots, L-1 \text{ and}
$$
$$
= \frac{1}{1 + \sum_{l=1}^{L-1} e^{\beta_k \mathbf{x}_p}}, k = L,
$$
(7)

where $f(k, p) = \beta_k \mathbf{x}_p$ is a linear predictor function, $\beta_k$ is the vector of the regression coefficient associated with damage category $k$, $\mathbf{x}_p$ is a vector the set of the explanatory variables associated with observation (stand) $p$ and $L$ is the number of the damage categories, here 3.

### 3.5. Support Vector Machine Method

Support Vector Machine method (SVM) is a machine learning technique presently actively adopted in remote sensing [23,51–54]. SVMs are supervised learning models with associated learning algorithms that analyze data used for classification or regression. Given a set of training examples, each marked as belonging to one or the other of two categories, an SVM training algorithm builds a model that assigns new examples to one category or the other, making it a non-probabilistic binary linear classifier [55].

SVMs are based on statistical learning theory and have the aim of determining the location of decision boundaries that produce the optimal separation of classes. [56]. In the case of a two-class pattern recognition problem with linearly separable classes, the SVM selects from among the infinite number of linear decision boundaries the one that minimizes the generalization error. Thus, the selected decision boundary will be the one that leaves the greatest margin between the two classes, where the margin is defined as the sum of the distances to the hyperplane from the closest points of the two classes [56]. The margin maximization is achieved using standard quadratic programming optimization techniques. The data points that are closest to the hyperplane are used to measure the margin and are referred to as support vectors.

If the two classes are not linearly separable, the SVM tries to find the hyperplane that maximizes the margin while, at the same time, minimizing a quantity proportional to the number of misclassification errors. The trade-off between margin and misclassification error is controlled by a user-defined constant [55]. SVM can also be extended to handle nonlinear decision surfaces by projecting the input data onto a high-dimensional feature space using kernel functions [56]. Radial basis functions with accordingly selected parameters are a typical choice to serve as kernel functions [51,52]. The gamma value varied here between 0.1 and 0.005 depending on the dataset.

As SVMs are designed for binary classification, this method appears to be an ideal fit for outlier detection problems, i.e. separating damaged forest class against intact using temporal dynamics of SAR backscatter. However, for estimating severity of damage (evaluating "change magnitude"), the approach is less suitable.

### 3.6. Area Estimates and Error Estimates

We used poststratified estimators for the area and area error estimators [57], as derived and suggested by Olofsson et al. [58]. The estimators use the confusion matrix and the area estimates of the categories based on an output map, that is, the pixel level estimates of the categories. The stratified estimators of the proportion of a category $k$ is

$$\hat{p}_{.k} = \sum_{i=1}^{L} W_i \frac{n_{ik}}{n_{i.}}, \tag{8}$$

where $n_{ik}$ is the $(i, k)$ element of the confusion matrix, observed counts on the columns and classified on the rows; $n_{i.}$ is the row sum of the row $i$; $L$ is the number of the categories; and $W_i$ is the proportion of the area mapped as category $i$. The area estimate of category $k$ is

$$\hat{A}_k = A \times \hat{p}_{.k}, \tag{9}$$

where $A$ is the total area mapped.

The standard error for the poststratified estimator of the proportion of area (Equation (8)) is estimated by

$$S(\hat{p}_k) = \sqrt{\sum_{i=1}^{L} \frac{W_i \hat{p}_{ik} - \hat{p}_{ik}^2}{n_{i.} - 1}}, \tag{10}$$

where $\hat{p}_{ik} = W_i \frac{n_{ik}}{n_{i.}}$, and standard error of the the area by

$$S(\hat{A}_k) = A \times S(\hat{p}_k). \tag{11}$$

An approximate 95% confidence interval is obtained as $A_k \pm 1.96 \times S(\hat{A}_k)$.

### 3.7. Confidence Intervals of Probabilities for Individual Observations Using ik-NN

The following procedure can be used to assess the uncertainty of the prediction of the damage and non-damage of individual stands or forest areas. The k-NN estimation and its improved version ik-NN produce probabilities for the predicted category on stand $p$. These probabilities can be calculated using the weights $w_{i,p}$ (Equation (4)) as follows

$$\widetilde{prob(k)}_p = P(\text{category}(p) = k) = \sum_{i \in I_p} w_{i,p} Ind_{(cat(i)=k)}, \tag{12}$$

where $k$ is the mode category based on the largest sum of the weights $w_{i,p}$ by categories on stand $i \in I_p$ and $Ind_{cat(i)=k}$ is an indicator function of the category in stand $i$. The confidence intervals for the probabilities of the mode for the individual stands were calculated using a linear model

$$\widetilde{prob(k)}_p = a + \sum_{l=1}^{n_f} b_l f_{l,p} + c \cdot k + \varepsilon, \tag{13}$$

where $f_{l,p}$ are the SAR features (Equation (5)); $k$ is the predicted damage category, a categorical variable (factor); $a$, $b$ and $c$ are the regression coefficients to be estimated ($b$ being a vector); and $\varepsilon$ is a normally distributed random error.

The confidence intervals of the predictions were calculated in a normal way using the estimator

$$\hat{V}_f = s^2 \, x_0 \, (X'X)^{-1} \, x_0' + s^2 \tag{14}$$

for the variance for the individual prediction with a predictor vector of $x_0$, residual sum of $s^2$ and design matrix $X$ consisting of the feature vectors $f$ and predicted categories $k$.

## 4. Results

### 4.1. Selection of the Data, Variables and Methods

The capability of the different features in damage area recognition was first studied (see Section 3.1). Other variables were some traditional stand level forest characteristics (Table 1) as well as slope, aspect and altitude calculated from the digital elevation model. The stand level variables tested were mean diameter, mean height, age, basal area and the volume of growing stock as well as volumes by tree species groups (Table 1).

The methods studied in windstorm damage recognition were the improved k-NN with feature optimization based on a genetic algorithm, called ik-NN method, multinomial logistic regression (MLR) and support vector machine (SVM) (Sections 3.3–3.5). The three models were used to classify the training data into the three categories, non-damaged, severely damaged and slightly damaged. The accuracies were validated using the validation data. The explanatory variables in the models were the SAR metrics calculated from the SAR images before and after the damage as well as the other characteristics mentioned above.

Each method has several parameters to control the performance of the method. Furthermore, the combinations of the explanatory variables is large. The number of the optional methods and variable combinations is thus really large. It was not possible to carry out all possible experiments. Only the results from a few tested combinations are reported here.

### 4.2. Selection of Basic Sentinel-1 Features

Selection between the intensity variables, average, median and quantiles were done using the SVM classifier and the Sentinel-1 scenes starting on 4 January 2017 and reaching

until 20 August, in total 19 image layers (see Table 4). The date of 20 August was selected close enough after the damage of 12 August. In total, three images after the damage were available. The averages of the intensities, more precisely, $\sigma^0$ (1), gave the largest OA in the validation data, 0.771 (Table 6). The averages also worked well in the cases of UA and PA. Only the results based on the averages are therefore reported here as the main results.

**Table 6.** The overall accuracies (OA) in training and validation data and user's accuracies (UA) and producer's accuracies (PA) by damage by categories in the validation data when using 19 Sentinel-1 cover (1 January 2017–20 August 2017) and the SVM classifier. A volume threshold of 75 m$^3$/ha is used in the data.

| Intensity | OA | | UA | | | PA | | |
|---|---|---|---|---|---|---|---|---|
| method | Training | Test | 1 [1] | 2 [2] | 3 [3] | 1 [1] | 2 [2] | 3 [3] |
| Average | 1.0 | 0.771 | 0.805 | 0.619 | 0.750 | 0.913 | 0.531 | 0.250 |
| Median | 1.0 | 0.747 | 0.840 | 0.511 | 0.412 | 0.913 | 0.531 | 0.250 |
| Quantiles | 1.0 | 0.763 | 0.788 | 0.647 | 0.625 | 0.930 | 0.449 | 0.208 |

[1] No damage, [2] Severe damage, [3] Slight damage.

### 4.3. Selecting a Time-Frame of Sentinel-1 Scenes

One goal of the study was to examine the minimum number of the scenes and the shortest time after the damage to achieve a feasible uncertainty level in windstorm detection. For this purpose: (a) only images until 20 August were used (19 images instead of the original 27 image layers); and (b) only images until 14 August were used, further limiting the number of images to 17. We also studied the importance of the scenes before the damage, that is, the achieved accuracies when some of the images acquired before the damage were left out of analysis. It turned out that 10 scenes gave almost as large accuracies as the entire set of the scenes from 4 January to 20 August 2017. The accuracies were thus calculated using the following numbers of the images: 27 (all scenes), 19 (all scenes until 20 August), 17 (all scenes until 14 August), the most important scenes until 20 August (10 scenes) and the most important scenes until 14 August (8 scenes) (see Section 4.4). Recall that the damage occurred on 12 August.

The importance of the Sentinel-1 scenes, acquired before the damage, was studied with all three methods. The date of these 10 scenes were: 16 January, 28 January, 9 February, 2 July, 14 July, 21 July, 26 July, 2 August, 19 August and 20 August. The dates of all available scenes in the period 4 January–20 August, in total 19 scenes, were 4 January, 16 January, 28 January, 9 February, 21 February, 2 July, 3 July, 8 July, 9 July, 14 July, 15 July, 21 July, 26 July, 2 August, 7 August, 8 August, 14 August, 19 August and 20 August (Tables 4 and 7).

When using all Sentinel-1 scenes from the period 4 January–14 August, as well as adding scenes cumulatively after 14 August, it was noticed that the overall accuracy (OA) increased only slightly. The maximum accuracy was obtained with the training and validation data with the scenes until either 12, 13 or 18 September, depending on the method. The OA in the validation data was 0.79 with SVM, 0.73 with ik-NN and 0.71 with MLR (Figure 8 and Table 7).

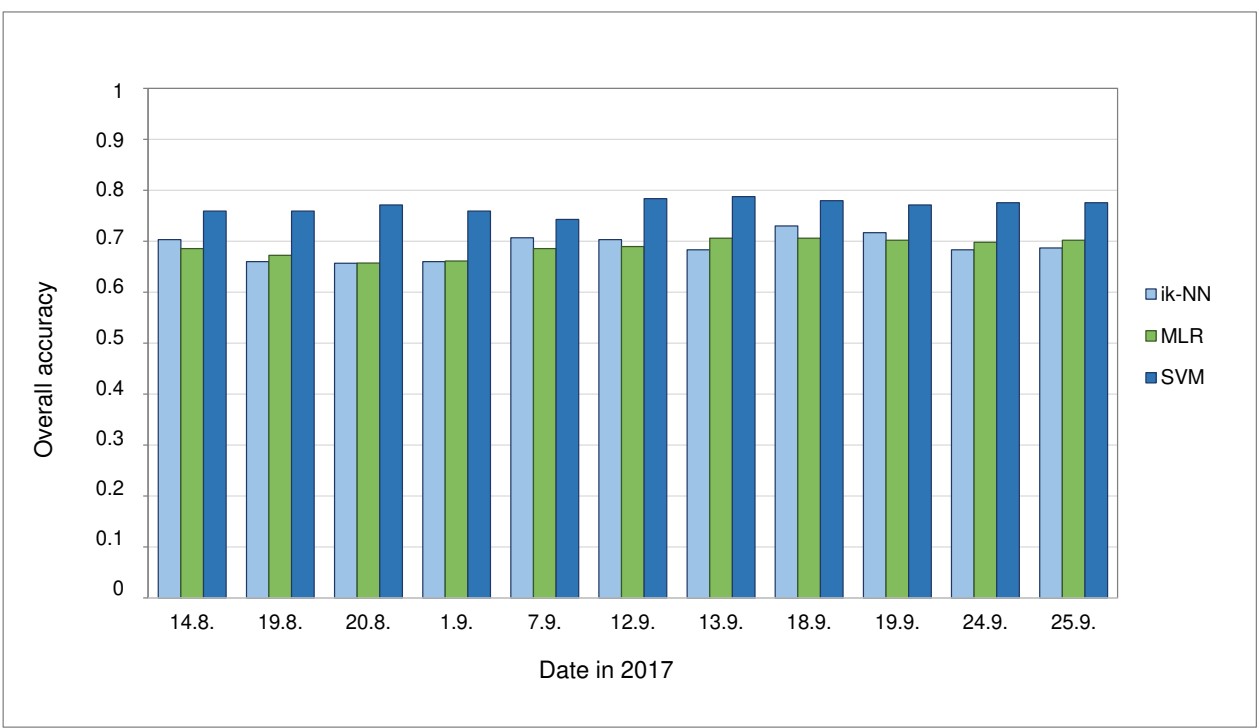

**Figure 8.** The overall accuracy with three different methods, ik-NN, MLR and SVM as a function of the acquisition date of the latest Sentinel-1 scene. The scenes were used until the date in the horizontal axis.

**Table 7.** The overall accuracy (OA), user's accuracy (UA) and producer's accuracy (PA) in the validation data using support vector machine (SVM), improved k-NN method (ik-NN) and multinomial logistic regression with four different Sentinel-1 datasets. The latest scene after the damage was from 20 August 2017, except for 27 scenes in which all Sentinel-1 scenes were used. Segmentation-based results are indicated with 'S'. A volume threshold of 75 $m^3$/ha was used for the data.

| Method and the Number of the Scenes | OA | UA by Category | | | PA by Category | | |
|---|---|---|---|---|---|---|---|
| | | 1 [1] | 2 [2] | 3 [3] | 1 [1] | 2 [2] | 3 [3] |
| SVM 8 scenes | 0.729 | 0.778 | 0.618 | 0.400 | 0.892 | 0.438 | 0.250 |
| SVM 10 scenes | 0.720 | 0.754 | 0.562 | 0.500 | 0.917 | 0.375 | 0.125 |
| SVM 17 scenes | 0.759 | 0.795 | 0.610 | 0.667 | 0.901 | 0.510 | 0.250 |
| SVM 19 scenes | 0.771 | 0.805 | 0.619 | 0.750 | 0.913 | 0.531 | 0.250 |
| SVM 27 scenes | 0.769 | 0.781 | 0.688 | 0.800 | 0.955 | 0.458 | 0.167 |
| SVM 8 scenes, S | 0.735 | 0.792 | 0.535 | 0.500 | 0.884 | 0.469 | 0.208 |
| SVM 10 scenes, S | 0.755 | 0.816 | 0.571 | 0.462 | 0.901 | 0.490 | 0.250 |
| SVM 19 scenes, S | 0.784 | 0.807 | 0.686 | 0.625 | 0.948 | 0.490 | 0.208 |
| SVM 27 scenes, S | 0.788 | 0.832 | 0.683 | 0.500 | 0.919 | 0.571 | 0.292 |
| ik-NN 8 scenes | 0.700 | 0.775 | 0.415 | 0.435 | 0.871 | 0.327 | 0.263 |
| ik-NN 10 scenes | 0.690 | 0.762 | 0.467 | 0.250 | 0.867 | 0.404 | 0.105 |
| Ik-NN 17 scenes | 0.703 | 0.781 | 0.432 | 0.434 | 0.867 | 0.365 | 0.263 |
| ik-NN 19 scenes | 0.630 | 0.747 | 0.286 | 0.133 | 0.814 | 0.308 | 0.053 |
| MLR 8 scenes | 0.703 | 0.742 | 0.435 | 0.583 | 0.917 | 0.208 | 0.292 |
| MLR 10 scenes | 0.712 | 0.763 | 0.464 | 0.533 | 0.904 | 0.271 | 0.333 |
| MLR 17 scenes | 0.686 | 0.771 | 0.407 | 0.348 | 0.879 | 0.229 | 0.333 |
| MLR 19 scenes | 0.657 | 0.794 | 0.326 | 0.296 | 0.808 | 0.286 | 0.333 |

[1] Damage, [2] Severe damage, [3] Slight damage.

### 4.4. The Accuracies with Different Methods

The overall accuracy of the classifications increases only slightly when adding the Sentinel-1 scenes after 1 or 2 scenes after the damage of 12 August (Figure 8) (see also Table 7). The latest date here is 20 August, except for 27 scenes in which all Sentinel-1

scenes were used. SVM gave the largest OA and ik-NN slightly larger than MLR. MLR is the only parametric method of the three methods tested. All methods presume the selection of the estimation parameters. The OAs are slightly larger when the backscattering-based features are calculated using segmentation-based $\sigma^0$ instead of stand boundaries-based ones (Table 7, notation 'S'). The proposed segmentation approach (tested only with SVM) increased the OA particularly when the number of the scenes is small (e.g., 10), as well as the PAs for the damage categories with 10 scenes. There are many possible combinations, therefore not all were tested here. The limited improvement can also be attributed to the fact that segmentation used the SAR data only now. A larger accuracy could be obtained when using also the inventory and other auxiliary data, similar to set of features used in the classification method.

### 4.5. The Damage Map Along with Area and Area Error Estimates

An example of the predicted damages in a map form was made using the ik-NN method. The damage was predicted only to forests in which the volume of growing stock exceeded 75 m$^3$/ha. Ten Sentinel-1 scenes with the dates of 16 January, 28 January, 9 February, 2 July, 14 July, 21 July, 26 July, 2 August, 19 August and 20 August were used.

A good practice with poststratified estimators was used in estimating the area of the damage categories (Section 3.6). The area of the classified forestry land (forest land, poorly productive forest land and unproductive land) was 799,000 ha. The severe damage was predicted for an area of 97,600 ha that corresponds to 12% of the mapped area and slight damages for an area of 76,200 ha (9.5% of the mapped forestry land area). Some individual pixels with wrong damage predictions increase the area of the damages. The large-scale damage patterns generally follow the windstorm patterns on 12 August 2017 (Figures 9 and 10). The poststratified estimates of the relative errors were 9% and 10%, respectively.

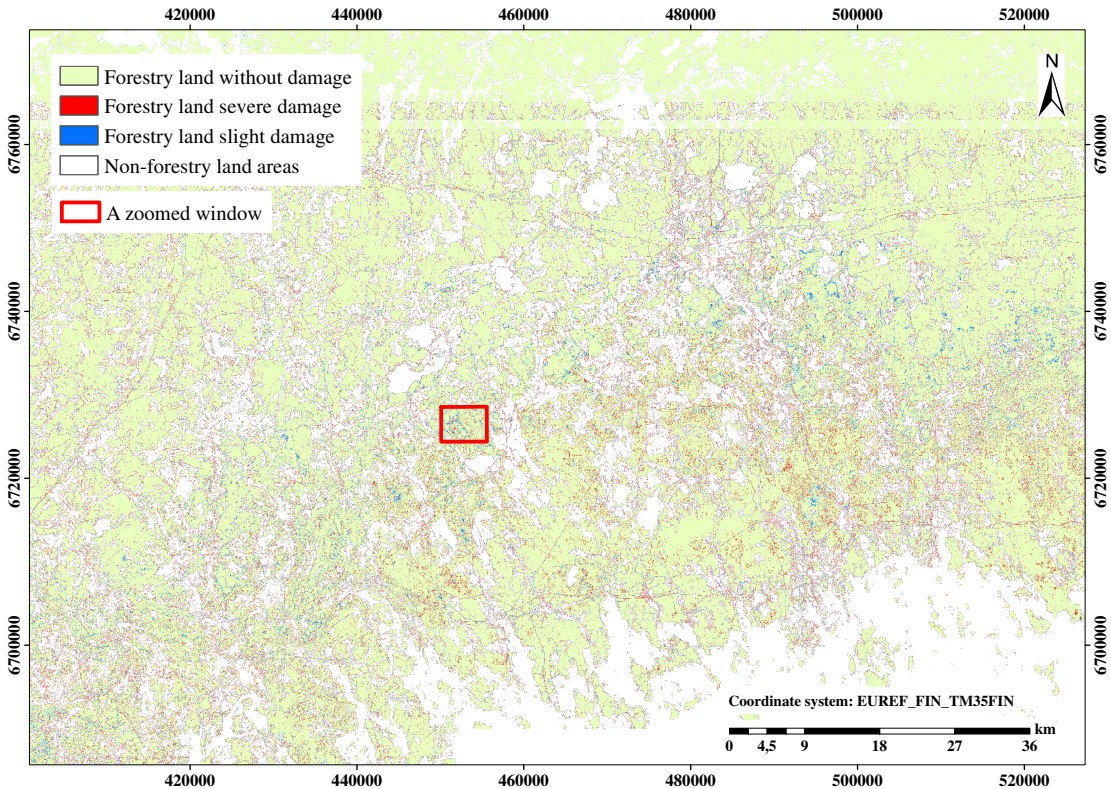

**Figure 9.** The predicted forest damages in the study area (EPGS:3067).

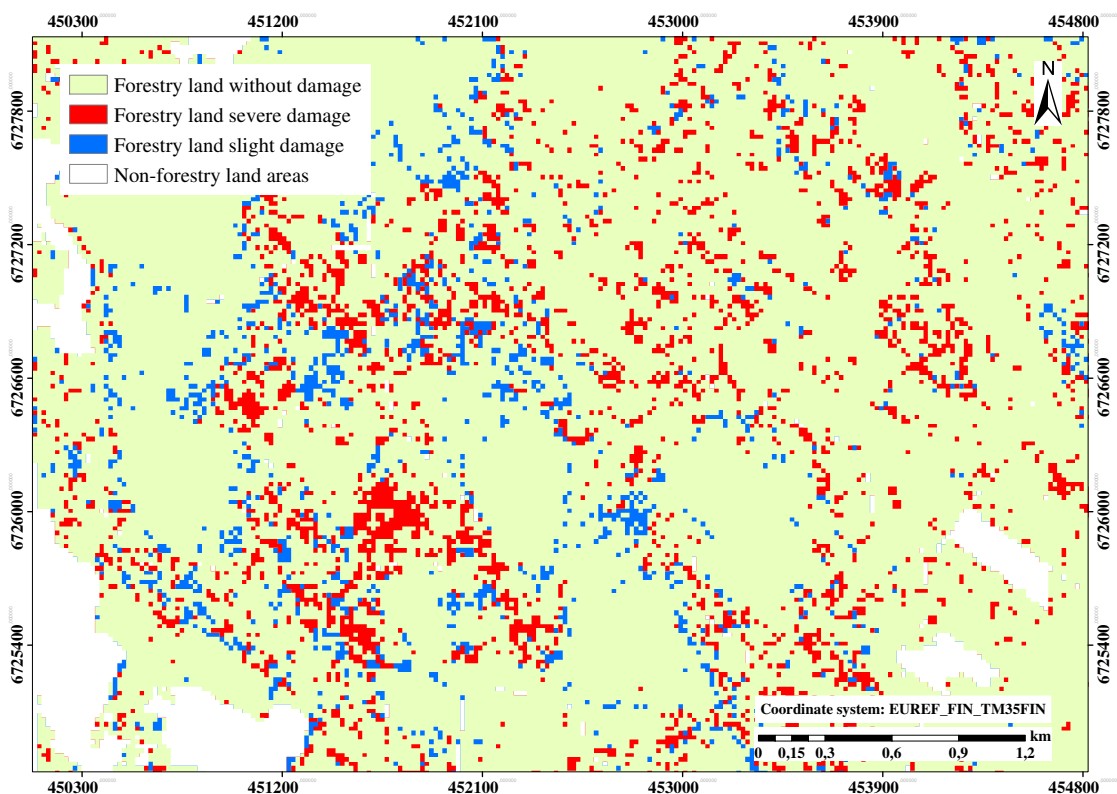

**Figure 10.** The predicted forest damages in a sub-area of the study area (EPGS:3067).

### 4.6. Confidence Intervals for Individual Stand Predictions

The probabilities of the predicted category in the validation datasets were assessed as given in Equation (12). The 95% confidence intervals of the predicted probabilities were calculated using Equation (14) and the Sentinel-1 datasets from the period 16 January–20 August, that is, in total 10 scenes (see Section 4.3). Basic statistics of the widths of the confidence intervals by the windstorm category are shown in Table 8. The median and mean of the intervals vary from 11% to 13%.

**Table 8.** Examples of the statistics of the 95% confidence intervals of the predictions of the probability of the category with the largest probability when ten Sentinel-1 scenes from the period 16 January–20 August 2017 were used. The statistics are shown by the threshold of the minimum of the stand area.

| Damage Category | Mean of Predictions | Std of Predictions | Min of Intervals | Median of Intervals | Mean of Intervals | Max of Intervals | Std of Intervals |
|---|---|---|---|---|---|---|---|
| 1 | 0.787 | 0.077 | 0.052 | 0.105 | 0.109 | 0.686 | 0.036 |
| 2 | 0.678 | 0.110 | 0.055 | 0.113 | 0.120 | 0.429 | 0.041 |
| 3 | 0.708 | 0.114 | 0.061 | 0.131 | 0.132 | 0.223 | 0.033 |

Figure 11 shows statistics of the widths of predictions intervals by the damage categories for the stands in the validation dataset together with the outliers. The statistics are the median, 25th and 75th percentiles and 1.5 times the interquartile range.

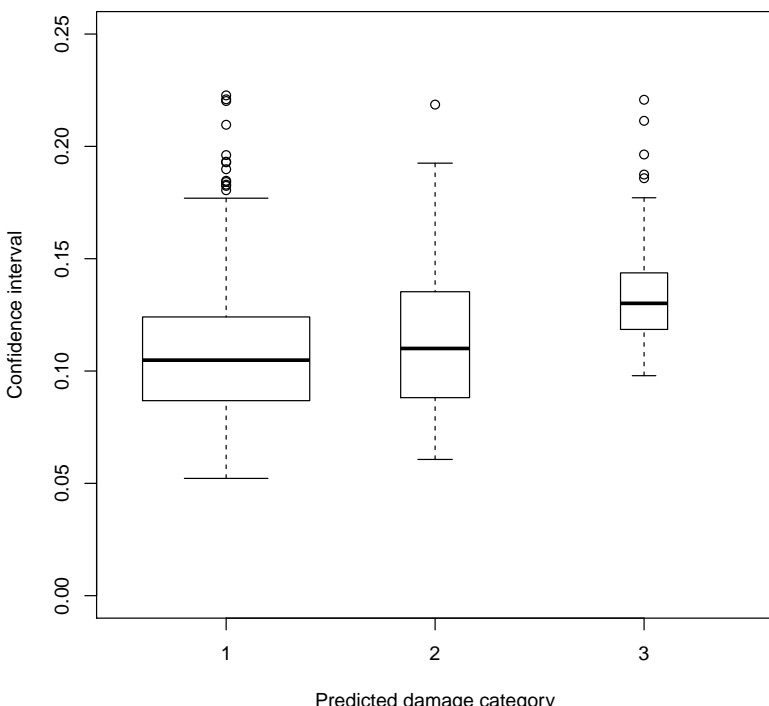

**Figure 11.** Distributions of the confidence intervals of the prediction probability of the category with the largest probability by the damage category, shown as boxplots, that is, median, 25th and 75th percentiles, 1.5 times of the interquartile ranges and individual outliers. The width of the box is proportional to the square root of the number of the observations of the damage category.

The accuracy of the individual prediction, that is, the confidence interval of the probability of the predicted category, also depends on the area of an individual damaged stand. The confidence interval becomes generally narrower when the area of a stand increases up to about 1–2 ha (Figure 12).

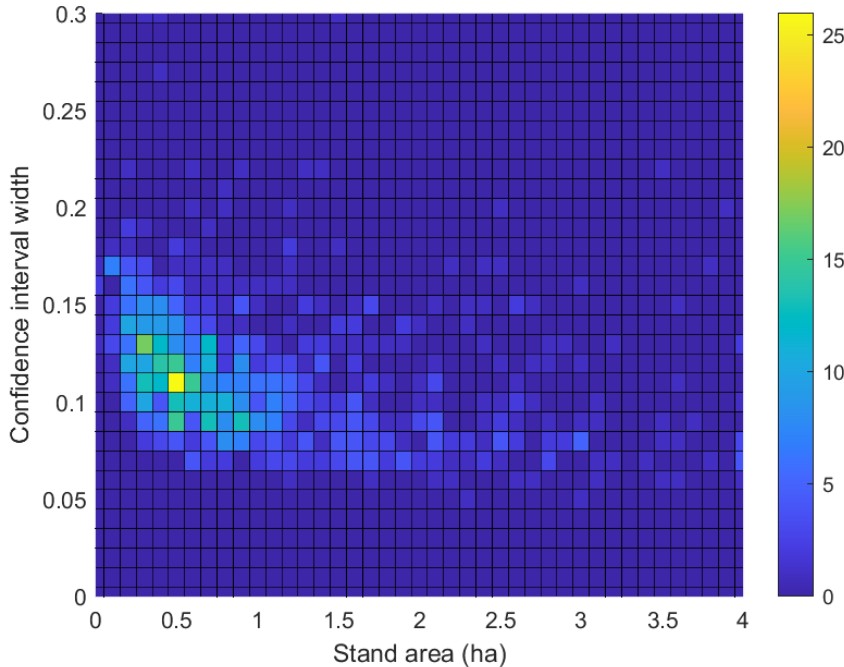

**Figure 12.** The width of the confidence interval of the probability of an individual prediction as a function of the area of the damaged stand.

*4.7. The Accuracies of the Methods without a Separate Training Data*

The use of specific training data is not necessarily possible in operational applications due to a need for very rapid or almost real-time localization of the damages. A final goal should be development of a method and a model that predict the locations of the damages without new training data and using only the existing information before the damage and SAR data (1–6 scenes) after the damage.

This possibility was tentatively tested in a simple way by selecting a set of the Sentinel-1 scenes and another distinct set for prediction. The cover numbers of the scenes for modeling were 2, 3, 4, 6, 7, 11, 12, 15, 16, 17, 18, 24, 25 and 26, and in prediction 1, 5, 8, 9, 10, 13, 14, 19, 20, 21, 22, 23 and 27 (see Table 4). The proportion of correctly predicted stands was 0.73. This simple test could already indicate if it is possible to develop an unsupervised damage prediction method.

## 5. Discussions and Future Work

The windstorm analyzed in this study took place during the Finnish summer, on 12 August 2017, wherefore the most important Sentinel-1 scenes were also from the summer. Earlier and other ongoing studies have revealed that late autumn or early winter is the best season for forest parameter estimation from SAR images in the boreal region (e.g., [59,60]). The results of this study show that the summer SAR images could also be applicable in forest change detection caused by a windstorm damage. One challenge when developing an operational damage monitoring method is that the windstorms occur all around the year, including during winter in Scandinavia and other boreal region. The weather conditions and the temperature can also rapidly change from above to below 0 °C. Under frozen conditions, a normalized radar cross-section decreases (e.g., [61]). Generally, freeze–thaw environmental transitions affect the classification methods and accuracies if they happen just after or during the damage and if a rapid assessment is needed.

This study provides some alternative methods to be developed further to be part of an operational windstorm damage monitoring system. Three different classification algorithm were tested to classify the forest observations on the three categories, non-damaged, severely damaged and slightly damaged forest patches. In total, 27 Sentinel-1 image covers were acquired and originally used, 16 before the damages and 11 after damages, in addition to the field observations and other geo-referenced data. The explanatory variables were derived from the intensities of the two polarizations VV and VH of the Sentinel-1 images and from the quantities of the other geo-referenced data. Two alternative analysis units were tested: (1) the forest stands or forest areas to be cut due to the windstorm damage; and (2) the forest patches constructed using a segmentation algorithm. The main analyses were carried out with the Alternative 1. The data were split to training data and validation for assessing the uncertainties of the results of the different methods. A statistical method was developed to construct the confidence intervals of the probabilities of the estimated damage categories. One goal was to find the minimum number of the images after the damages for a rapid operational monitoring method.

Using calculation units that are derived with a segmentation algorithm, that is, the units that are homogeneous with respect to backscattering coefficient, slightly increased the overall accuracies (OAs), and in some cases also the user's accuracies (UAs). In some cases, the UAs and producer's accuracies (PAs) were smaller than with the given boundaries. This may imply that the given boundaries followed the damaged areas or the effect of variation in the image conditions on the results is significant or the segmentation-based approach needs further development.

Although some windstorm damage studies have been published so far with space-borne SAR data, quantitative uncertainty assessments are generally lacking. Many comparisons with accuracy figures are thus not possible. Our results are competitive with, e.g., those of Thiele et al. [21]. Our study showed that the damages could be identified even a few days after the damage, which is quite unique. On the other hand, we should keep in mind that windstorm damages vary by the areal extent of the damaged forest patches and

also by severity. Furthermore, forest structures and imaging conditions vary wherefore uncertainty quantities are not necessarily comparable.

Preliminary tests showed that it could also be possible to develop an unsupervised method for windstorm damage monitoring, that is, to detect the changes without a specific training data.

Detection of the windstorm damage is a demanding task. Severity of the damages changes within the area of one storm, even in a relatively small area, e.g., the one in this study, 100 km × 100 km, and depends on many factors, e.g., the structure of the growing stock, the soil properties, the terrain elevation variation and the small scale spatial variation of growing stock. Forests next to an open area or a young forest are more vulnerable to damages than the forests surrounded by mature forests. Furthermore, in addition to the changes in the growing stock, many other factors affect the changes of backscatter and also in a short time interval, e.g., changes in the moisture of the tree canopies and soil. The possibility to frequently acquire SAR data is thus important.

It should be recognized that further work is needed for a near-real-time operational monitoring system.

In the continuation work, the accuracy of the estimates will be improved by further method development and additionally using interferometric SAR data as well as meteorological data. The use of other geo-referenced data, such as land-use data, forest age and soil data and forest data from the surrounding areas, may improve the classification accuracy because the windstorm damages occur often on the borders of open areas, newly constructed roads and power-lines as well as next to young forests or forest regeneration areas.

Potential of interferometric SAR coherence, possibly combined with backscatter intensity information, should be studied using Sentinel-1 multitemporal imagery, even though temporal decorrelation can limit its utility [62]. Further, time series of bistatic TanDEM-X scenes can be used for mapping forest change, due to high sensitivity to the vertical structure of the forests [63,64]. For the latter, the limiting factor is data availability over large areas with small latency.

Coming and existing satellite missions and constellations with frequent and tailored data acquisition increase the availability of data. It is also important that data providers adopt a systemic data acquisition strategy similar to Sentinel-1 and ALOS/ALOS-2 missions in connection with the hazard monitoring, particularly windstorm detection. A background mission with at least seasonal global coverage can be suitable.

## 6. Conclusions

The methods to localize the forest damages caused by windstorms using space-borne SAR data were developed and possibilities to an operative system investigated. Multitemporal Sentinel-1 time series were used.

Support vector machine (SVM) gave the largest overall accuracies among the three methods tested, improved k-NN (ik-NN), multiple logistic regression (MLR) and SVM. The proportion of correctly classified stands (OA) in a separate validation data was 79%, and 75% if only one Sentinel-1 scene after the damage was used. The user's accuracy (UA) for severe damages was 62%, and 75% for slight damages. The producer's accuracies (PAs) were somewhat lower. The accuracy of 75% was achieved using only one Sentinel-1 scene after the damage, here two days after the damage, in addition to the data before the damage.

Using segmentation-based calculation units only slightly increased the OA, implying that this approach may presume further work. Most likely, not only SAR data, but also inventory and other auxiliary data should be used in the segmentation methodology.

The study indicates that the damages could be localized using only one Sentinel-1 scene after the damage implying a time-lag of potential satellite SAR-based assessment method would be just a few days after the damage. This gives promises that a SAR-based near-real-time semi-automatic operative system to monitor windstorm damages is feasible.

**Author Contributions:** Conceptualization, E.T., J.P., O.A., H.H. and G.R.; methodology, E.T.; validation, E.T. and H.H.; formal analysis, E.T.; investigation, E.T., O.A. and J.P.; data curation, G.R., H.H. and E.T.; writing—original draft preparation, E.T.; writing—review and editing, E.T., O.A. and J.P.; visualization, E.T., G.R and O.A.; and project administration, E.T. and J.P. All authors have read and agreed to the published version of the manuscript.

**Funding:** The study was funded by Business Finland as a part of the project conducted by the Finnish Forest Centre, grant number 4480/31/2018. O.A. was supported by Business Finland MULTICO project under grant 6501/31/2019. All authors were also supported by their Institutes.

**Institutional Review Board Statement:** Not applicable.

**Informed Consent Statement:** Not applicable.

**Data Availability Statement:** Sentinel-1 satellite data are available from ESA via Copernicus Open Access Hub at no cost. Derived data supporting the findings of this study are available from the corresponding author E.T. on request.

**Acknowledgments:** The damage and non-damage stand data with stand boundaries were provided by the Finnish Forest Centre. Four anonymous reviewers provided constructive comments that improved the text. We thank all institutions and individuals who have contributed to the article.

**Conflicts of Interest:** The authors declare no conflict of interest.

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
