# Peer review of "Detection of Forest Windstorm Damages with Multitemporal SAR Data—A Case Study: Finland"

_remotesensing, doi:10.3390/rs13030383_

Round 1

Reviewer 1 Report

The manuscript under review concerns the use of SAR remote sensing to identify areas damaged or destroyed by windstorms in Finland.
Due to climate change and the increasing frequency of extreme phenomena, and the lack of adaptation of forest ecosystems to new phenomena, the analyzed topic is particularly important. The topic is also close to my interests in the field of forest degradation and the use of remote sensing to assess damage.
Overall, I evaluate the manuscript positively, with a good perspective for publication. However, the authors did not avoid errors in the text, preparation and presentation of the results at the same time. They do not necessarily have to be big mistakes, but often oversights or omissions.

I rate the manuscript as minor revision. It is necessary to take into account comments and correct the text.

Below are my comments:

Editing the title
Add 'case study: Finland'

Abstract
Add 2-3 sentences about the intensity of windstorm damages

Line 30
double keywords

Introduction
requires improvement, supplementation, expansion, especially in the background
there is no in-depth review of the literature on windstorm forest damages and monitoring methods

Lines 42-46
Give references

Lines 50-54
Give references
It is not enough to list the available methods
I also don't like the practice of writing that someone else has already done a review. Someone else is not you and this is not his text

Line 60-61
What implies that the SAR is the best?

Line 86-87
I am against writing that there were only a few other studies, how do you know that only a few, maybe you didn't reach the other dozen? and besides, if you write that only a few, why do you reference only one?

Line 87,
why is a double bracket? ([])? similar in other places
Either there is ([]) or only [] - where these changes come from

Material
General arrangement of figures and tables in the text in relation to their reference in the text - figures and tables are too far from the place where they are described, it makes it very bad to analyze the text

Line 171
Describe the meteorological event, windstorm

Line 173
Describe the forest stands more precisely

Line 174-175
I think some punctuation marks are missing

Lines 184-187
Please also characterize forest habitats, types of these stands, predominant species, age class

Table 4
A year is missing

Figure 2
The graphics are too small, the stands location is unreadable, enlarge the maps,
It seems to me that showing VV and VH polarization at the same time is unnecessary, because background is unreadable at this size
Define a coordinate system with the EPSG code
there is no north arrow
What is the white rectangle in the upper right corner?
In the legend, describe the red frame as the enlarged area in Figure 3

Figure 3, 4
Similar comments,
I think one polarization is enough
Deficiencies as in Fig. 2
Why are the dates different from Fig. 2?

Figure 5
Describe on the map in the legend what these values are - volume of the growning stock
Mark the red frame in the legend

Methods
Don't start the chapter with figure, why is figure 6 here?

Figure 6
Notes as before

Chapter 3.1 / 3.2.1
Give references to the method as you did to the next methods

Lines 276-284
Maybe there is another way to describe it, e.g. a table or a scatter plot? it is illegible in the text

Results
Table 5, 6
Shorten the title table, give the parameter descriptions below the table

Figure 9, 10
Zoom in
In Figure 9, mark the red box in the legend
Other remarks as above

Fig 12
Describe the color scale in the figure

In the results and discussion, I miss a clear indication of the research results. Summaries of the whole, statement of results, main assumptions of the method, parameters of the analysis and results. Complete it, it will be such a gem at the end.

Reviewer 3 Report

The study presents a unique approach to using SAR data in mapping windthrows. Several proposed techniques are presented and tested using field data available in large areas. 

Comments

Figure 2 – what is the red square meaning. You should change the scale of the maps, is hard to see the differences between the polarization

Figure 3 and 4 are repeating – to many visual examples. If you add the detailed map on a better scale in figure 2 you can join all figures (2,3,4) in one

Figure 5 – is hard to see inside the red square please add the detail from figure 6 and join them into a single figure

Methods section – is hard to identify the software used in the processing, please give a reference in text or even better in figure 7

Sections 3.3, 3.4 , 3.5 and 3.6 need to be shortened – most of the material is theoretical and they can be moved in the auxiliary section or simplified a lot

Please address the limitation of using a volume threshold of 75 m3/ha used in the data

Reviewer 4 Report

Just two remarks:

1.Expressions for standard deviations in eqs 2 and 10 are incorrect.

2.An influence of forests NRCS seasonal variations because of the wood freezing/thawing on the quality of the damaged forests discrimination is desirable to be discussed.

Round 2

Reviewer 2 Report

Please check the Table7. Below the table, 4, 5 and 6 indicate no damage, servere damage and slight damage?

Author Response

We thank the reviewer for the very thorough work.

Our response is below.

Comments and Suggestions for Authors
Please check the Table 7. Below the table, 4, 5 and 6 indicate no damage,
severe damage and slight damage?

Response:
Thank you for picking up these typos.
In fact, LaTeX overleaf has numbered the footnotes
continuously running when using our original commands.
We corrected.
The correction is marked as purple.